# Limits of Applicability of the Composite Fermion Model

**DOI:** 10.3390/ma14154267

**Published:** 2021-07-30

**Authors:** Janusz E. Jacak

**Affiliations:** Department of Quantum Technologies, Wrocław University of Science and Technology, Wyb. Wyspiańskiego 27, 50-370 Wrocław, Poland; janusz.jacak@pwr.edu.pl

**Keywords:** Laughlin function, FQHE, composite fermions, GaAs, graphene monolayer and bilayer

## Abstract

The popular model of composite fermions, proposed in order to rationalize FQHE, were insufficient in view of recent experimental observations in graphene monolayer and bilayer, in higher Landau levels in GaAs and in so-called enigmatic FQHE states in the lowest Landau level of GaAs. The specific FQHE hierarchy in double Hall systems of GaAs 2DES and graphene also cannot be explained in the framework of composite fermions. We identify the limits of the usability of the composite fermion model by means of topological methods, which elucidate the phenomenological assumptions in composite fermion structure and admit further development of FQHE understanding. We demonstrate how to generalize these ideas in order to explain experimentally observed FQHE phenomena, going beyond the explanation ability of the conventional composite fermion model.

## 1. Introduction

The fractional quantum Hall effect (FQHE) was discovered experimentally in 1982 by Tsui, Stormer and Gossard in GaAs 2DES [1], shortly after the experimental discovery of the integer quantum Hall effect (IQHE) in 1980 by von Klitzing in a 2D electron system in a perpendicular magnetic field [2]. There was great interest in both of these effects caused because they revealed an unexpected quantum behavior that goes beyond the conventional quantum mechanics theory. The experimental achievements have been distinguished with the Nobel prizes (for IQHE in 1985 and for FQHE in 1998) and have opened a wide discipline of condensed matter related to Hall physics in 2D systems, including graphene (especially after the Nobel prize for graphene discovery and its description in 2010) and topological insulators [3], study of which rapidly flourished in the past few years.

Nevertheless, despite intensive development of the relevant theory, the understanding of FQHE is still not complete and apparently cannot be achieved in the framework of local quantum mechanics. Various routes towards explanation of queer discrete structure of FQHE hierarchy have been proposed, like the multiparticle wave functions for the corresponding correlated states suggested by Laughlin in 1983 [4] (also honored by the Nobel prize in 1998), the hierarchy of daughter generations of anyons by Halperin and Haldane [5,6,7], multi-component systems by Halperin [8] or composite fermions (CFs) by Jain and Wilczek [9,10]. The Laughlin function, although proposed without a derivation in a phenomenological manner [4,11], was the most illuminating theoretical discovery and gave rise to various phenomenological models intended to decipher the physics behind correlations of electrons at FQHE. This multiparticle wave function proposed by Laughlin for *N* interacting electrons on the plane exposed to a strong quantizing perpendicular magnetic field was the simple generalization of the Slater function for *N* noninteracting 2D electrons at the field $B=B_{0}$, at which the degeneracy of Landau levels (LLs) $N_{0}\left(B_{0}\right)=N$, where $N_{0}\left(B\right)=\frac{BSe}{h}$ is LL degeneracy, *S* is the 2D sample surface, *h* is the Planck constant and *e* electron charge. This Slater function must be built of *N* various single-electron states in the lowest LL (LLL) and the rotation symmetrical gauge of the magnetic field attains the form of the Vandermonde determinant $\prod_{i>j}^{N}\left(z_{i}-z_{j}\right)$ ($z_{i}$ is the coordinate of *i*-th electron on the plane represented as the complex number) multiplied by the envelope function $e^{-\sum_{i}^{N}{\left|z_{i}\right|}^{2}/4l_{B}^{2}}$, where $l_{B}=\sqrt{\frac{\hbar }{eB}}$ is so-called magnetic length [12,13]. Laughlin exchanged only the Vandermonde polynomial for the Jastrow polynomial $\prod_{i>j}^{N}{\left(z_{i}-z_{j}\right)}^{q}$, *q*—odd integer, and demonstrated numerically that the resulted function pretty well approximates the exact ground states of *N* interacting electrons on the plane *S* at the magnetic field $B=qB_{0}$, i.e., at fractional filling of the LLL, $\nu =\frac{N}{N_{0}\left(B\right)}=\frac{1}{q}$ [4,11,14].

The Laughlin function has been postulated and not derived. It has been proved that it is impossible to derive this function within local quantum mechanics, and topological methods are required [15]. This is connected with the fact that the FQHE collective state is not a conventional phase. It is not characterized by any local order parameter and not associated with any symmetry breaking. Instead, FQHE phases are topologically correlated multielectron states with long-range quantum entanglement of all electrons simultaneously [16] protected by homotopy invariants [15]. The reason for FQHE correlations is the Coulomb interaction of 2D electrons at magnetic field presence but not due to a binary scattering of electrons in any coherent channel. The Coulomb interaction induces in 2D Hall systems the topological invariants which protect particular homotopy phases of FQHE [15] in a universal manner, independently of microscopic differences in various experimental systems, no matter in GaAs, in graphene and even in topological Chern insulators with magnetic field substituted by the Berry field [3].

In the present paper we will derive with mathematical rigor in the framework of topological approach the model of CF, which was usable for the main part of the FQHE hierarchy in the LLL of GaAs 2DES. We will demonstrate explicitly why the CF model failed in explanation of the whole hierarchy of FQHE, in particular of so-called enigmatic states in the LLL of GaAs, of fractional Hall hierarchy in higher LLs in GaAs [17,18] and in graphene [19,20] (especially in bilayer graphene [21,22,23] or bilayer GaAs [24,25]). We also will show how to generalize the CF model.

## 2. Laughlin Function and CF Model

The proposition by Laughlin [4] of his famous multiparticle wave function for FQHE resolved itself to the substitution of the Vandermonde polynomial in the Slater determinant by the Jastrow polynomial, i.e., instead of the Slater function for completely filled LLL of gaseous system of *N* electrons on the surface *S* at field $B=B_{0}$,
(1)$$\begin{array}{c} \Psi \left(z_{1},\ldots ,z_{N}\right)=\mathrm{const}.\left|\begin{array}{cccc} 1 & z_{1} & \ldots & z_{1}^{N-1}\\ 1 & z_{2} & \ldots & z_{2}^{N-1}\\ \ldots & \ldots & \ldots & \ldots \\ 1 & z_{N} & \ldots & z_{N}^{N-1} \end{array}\right|e^{-\sum_{i=1}^{N}{\left|z_{i}\right|}^{2}/4l_{B}^{2}}\\ =const.\prod_{i>j}^{N}\left(z_{i}-z_{j}\right)e^{-\sum_{i=1}^{N}{\left|z_{i}\right|}^{2}/4l_{B}^{2}}, \end{array}$$

Laughlin suggested the function for interacting *N* electrons at fractional filling $\nu =\frac{N}{N_{0}\left(B\right)}=\frac{1}{q}$ (where q=2k+1, k=1,2,3,…, $N_{0}\left(B\right)=\frac{BSe}{h}$ is the degeneracy of LLs), i.e., at field $B=qB_{0}$,
(2)$$\Phi \left(z_{1},\ldots ,z_{N}\right)=const.\prod_{i>j}^{N}{\left(z_{i}-z_{j}\right)}^{q}e^{-\sum_{i=1}^{N}{\left|z_{i}\right|}^{2}/4l_{B}^{2}},$$
where $l_{B}=\sqrt{\frac{\hbar }{eB}}$. The function (Equation 2) is not in Slater determinant form anymore, but still is an antisymmetric function as *q* is an odd integer. However, the phase shift caused by the exchange of the *i*-th particle with the *j*-th one is qπ and not π as in the case of function (Equation 1)—this is the difference known as a Laughlin correlation. Both functions (Equation 1) and (Equation 2) are completely quantumly entangled (they reveal the long-range entanglement of all electrons in the system simultaneously), i.e., they cannot be separated to a product form in tensor Hilbert space $H=H_{1}⊗\ldots ⊗H_{N}$ where $H_{i}$ is the single-electron Hilbert space for *i*-th electron.

Even though the modification in the Laughlin function is quite simple, the physics behind the Laughlin correlations (the phase shift qπ when particles interchange) is not fully understood as of yet. The most popular trial to elucidate this problem was the CF model [9,26]. It is a phenomenological model using an artificial construction of the hypothetical composite particle consisting of an electron and attached to an even number of flux quanta $\frac{h}{e}$ of some fictitious magnetic field. The origin of this field is not specified as the manner of attaching its flux quanta to individual electrons. Such a construction allows, however, for an additional phase gain when two composite electrons mutually interchange their positions by virtue of the Aharonov-Bohm-type effect [10,27]—cf. Figure 1. Indeed, if two *e*-charged fermions (electrons) with pinned fluxes ϕ mutually exchange their positions on the plane, the multiparticle wave function of both gains the phase shift π+ϕe/2ℏ [10], and for $\phi =2k\frac{h}{e}$ one gets the phase shift π+2kπ=qπ in agreement with Laughlin correlations (cf. Figure 1 central panel). Authors of the CF model supposed that the dressing of electrons with magnetic field flux quanta is originated in the Coulomb interaction of electrons analogously to Landau quasiparticles in metals [26]. This is, however, impossible, as the mass operator for Coulomb interacting electrons cannot produce a magnetic field in any channel of electron scattering [28], which precludes the concept of composite fermions as Landau type quasiparticles. Apparently CFs are a phenomenological illustration of another physical mechanism out of the framework of local quantum mechanics and Dyson equation type perturbative treatment of the Coulomb interaction of electrons [28,29].

The most important further assumption in the CF model was the mapping of FQHE states in the LLL of GaAs at field *B* onto IQHE at complete fillings of the auxiliary spinless gaseous Hall system at magnetic fields reduced by the mean value of the auxiliary fictitious magnetic field, which fluxes were pinned to electrons to create CFs. This assumption also has no formal justification but correctly reproduces the main line for FQHE in the LLL in GaAs both in its spin-up and spin-down sublevels [9], in the following form,
(3)$$\nu_{CF}=\frac{y}{(q-1)y\pm 1},$$
where q−1 is the number of fictitious magnetic field flux quanta pinned to electrons (q=2k+1 is an odd integer) and y=1,2,3,… is the number of completely filled LLs in the auxiliary *spinless* gaseous 2D Hall system, and ± corresponds to the compatible or opposite direction of the resultant field (the external field *B* reduced by the mean field of fluxes attached to electrons in construction of CFs) with respect to the external field *B*. Equation (Equation 3) defines the main part of the FQHE hierarchy in the spin-up (with respect to the field *B*) subband of the LLL and, if shifted by one, in the spin-down subband.

However, not all fractions for FQHE observed experimentally in the LLL of GaAs [30] belong to the hierarchy (Equation 3). The series of these additional fractions like $\nu =\frac{4}{11},\;\frac{5}{13},\;\frac{3}{8},\;\frac{3}{10},\ldots$ (cf. fractions marked in color in Figure 2) correspond to FQHE states frequently referred to as enigmatic. They are out of reach for the conventional CF model, and, moreover, they are assisted with typical for FQHE states plateaus in $R_{xy}=\frac{h}{e^{2}\nu }$, but local minima in $R_{xx}$ at these states are nonzero in contrary to zero longitudinal resistivity at states from the hierarchy (Equation 3)—cf. Figure 2. Neither the CF model, nor other conventional theories of FQHE, are able to elucidate this latter behavior.

In the model of CFs the next assumption concerns a trial wave function for FQHE state from hierarchy (Equation 3) in the form of the gaseous wave function in *y*-th spinless LLs fully filled in resultant magnetic field for CFs and ’projected’ onto the LLL. The latter procedure is required to avoid singularities in wave functions in higher LLs, which cannot be present in the multielectron wave function at the arbitrary fractional filling rate of the LLL, where even in the interacting system the trial wave function must be holomorphic without any poles (analytical in the whole domain). The procedure of ‘projection’ onto LLL is, however, not unambiguously defined. It plays the role of the variational procedure to minimize the averaged energy over trial wave function [26].

All above listed assumptions in the CF model are not supported by any formal derivation and, moreover, the so-called multicomponent wave functions by Halperin [8] for the FQHE states approximate the activation energy of these states equally well as compared to energies assessed by exact diagonalization of interaction in small models and determined from experimental measurements. In the following paragraph we will present a topological approach to FQHE correlations in which it is possible to rationalize the CF model and to explain how CF model is relatively effective at fillings rates from the hierarchy (Equation 3). The nonlocal topological approach allows us to also develop the theory for cases out of reach of conventional CF formulation. We will demonstrate that the role of the interaction of electrons in FQHE is not perturbative and therefore none local order parameters related to some mass operator and spontaneously broken symmetry can be defined for quantum phases at FQHE.

## 3. Topological Invariants in 2D Systems of Interacting Electrons in Magnetic Field

Topological approach to FQHE [15,31,32] is a nonlocal theory which identifies topological homotopy-type invariants protecting specific to 2D collective quantum phases of interacting electrons exposed to the perpendicular strong magnetic field. The central notions are here cyclotron braid subgroups, which define various patterns of correlation of Coulomb repulsing electrons deposited on 2D uniform positive jellium in the presence of quantizing magnetic field (or Berry field in the case of Chern topological insulators [3]). The cyclotron braids can be defined only in the case of perfect spatial commensurability of cyclotron orbit size and related cyclotron braids with nearest or next-nearest electrons distributed according to Wigner crystal of electrons at T=0 K (cf. Appendix A). As the Wigner crystal can be defined only for repulsing electrons thus the homotopy invariants related to this commensurability condition can exist exclusively in the interacting systems and not in a gaseous one. The specific structure of finite sized 2D cyclotron braids multiloop their form in agreement with the algebraic group properties and the size of braids with additional loops is greater than loopless ones in the case when the loopless ones are too short to match the closest electrons in the Wigner crystal [15,31]. The proof of this surprising fact is done by application of the Bohr-Sommerfeld rule to multiply connected configuration space [32] (cf. Appendix B).

Taking into account the possibility of realization multiloop cyclotron orbits and the accommodation of their loops to nearest or next-nearest neighbors of various rank in electron classical Wigner crystal at T=0 K, one obtains the most general hierarchy of FQHE states in the LLL protected by the following homotopy invariants [15,33],
(4)$$\frac{BS}{N}=\frac{h}{x_{1}e}\pm \frac{h}{x_{2}e}\pm \ldots \pm \frac{h}{x_{q}e},$$
where $\frac{h}{e}$ is the fundamental quantum of magnetic field flux, $\frac{BS}{N}$ is the flux of external magnetic field *B* per one particle in 2D system with *N* particles and surface *S*, q=2k+1 is odd number of loops in multiloop cyclotron orbit (cyclotron braids are half-pieces of cyclotron orbits and have *k* loops; in the simplest case such a *k*-loop cyclotron braid $\sigma_{i}^{2k+1}$ is the exchange of particles *i*-th and (i+1)-th ones, similarly as $\sigma_{i}$ generator—cf. Figure 3 and Figure 4). Plus/minus before components in the sum (Equation 4) correspond to compatible or contrary circulations of consecutive loops corresponding to the components in sum (Equation 4). We see that (Equation 4) is the decomposition of the external magnetic field *B* flux per one electron into the contributions of all loops of multiloop cyclotron orbit (q=2k+1-loop in this case—it must be odd integer because half of it is a braid with additional *k* loops and it has to describe exchanges of *i*-th electron with (i+1)-th one similar to loopless simplest braid $\sigma_{i}$ [15,31], cf. illustration in Figure 4). Each loop in this decomposition must perfectly fit to nearest or next-nearest neighbors in the Wigner crystal of electrons. The integer factors $x_{i}$ define fractions of electrons $\frac{N}{x_{i}}$ corresponding to fractions of next-nearest neighbors of various rank in the Wigner crystal, cf. Figure 5 and Appendix A. Thus, $x_{i}=1$ refers to nearest neighbors, $x_{i}=2,\;3,\ldots$—to next-nearest neighbors sublattices in Wigner crystal of consecutive rank. The decomposition (Equation 4) is possible only in 2D where all loops of the multiloop cyclotron orbit share the same surface. The amount of the flux contribution of a single loop is $\frac{h}{e}$ (the fundamental flux quantum, cf. Appendix B) and for N/x electrons in some sublattice of next-nearest neighbors in the Wigner crystal a single loop commensurability condition is $\frac{BS}{N/x}=\frac{h}{e}$, which explains the form of components in (Equation 4) with $x_{i}$ in denominators. The filling rates corresponding to the homotopy invariants (Equation 4) equal to,
(5)$$\nu =\frac{N}{N_{0}}={\left(\frac{1}{x_{1}}\pm \frac{1}{x_{2}}\pm \ldots \pm \frac{1}{x_{q}}\right)}^{-1},$$
because the degeneracy of LLs is $N_{0}=\frac{BSe}{h}$.

If $x_{1}=\ldots =x_{q-1}=1$ and $x_{q}=y$ then the filling rate hierarchy (Equation 5) gives the CF hierarchy (Equation 3) (maintaining plus/minus before only last term in (Equation 5)). In the case of additionally $x_{q}=y=1$, (Equation 5) reproduces the hierarchy of FQHE defined by the Laughlin functions at $\nu =\frac{1}{q}$. We see thus that the general homotopy invariants admit the CF illustration of all loops of the multiloop cyclotron orbit fit to nearest neighboring electrons in the Wigner crystal except for one loop, which fits with next-nearest neighbor sublattice containing $\frac{N}{y}$ electrons. The q−1 components with $x_{i}=1$ defining the commensurability of q−1 loops with the closest neighbors are artificially illustrated in the CF model in addition to electrons q−1 flux quanta of a fictitious magnetic field (Figure 1). In fact such field does not exist and fluxes $\frac{h}{x_{i}e}=\frac{h}{e}$ for $x_{i}=1$ are contributions of particular loops to the external magnetic field flux per one electron $\frac{BS}{N}$ in accordance with formula (Equation 4). The single parameter of freedom $y=x_{q}$ in such a special case of the general invariant (Equation 4) was erroneously associated with *y*-th spinless auxiliary LL in the model of CFs. At complete filling of this *y*-th spinless LL it contains only $\frac{N}{y}$ electrons, i.e., this fraction of electrons coincides with the amount of appropriate rank next-nearest neighbors in the Wigner lattice (cf. Figure 5). This accidental coincidence of two integers—fraction of next-nearest neighbors in Wigner lattice and the number of consecutive spinless LLs— is a chance to approximate the true wave function by the wave function of auxiliary spinless *y*-th LL adjusted in a variational manner to the LLL by projection onto LLL in the CF model [26]. Such a construction of the wave function in the CF model is conceptually false (it is not only an interpretation) and can be treated at most as the heuristic variational procedure for searching a trial wave function for a particular state of FQHE in the LLL. The true eigen function for this state must be a multiparticle function, which transforms in agreement with scalar unitary representation of the cyclotron braid unitary subgroup and must be of homogeneous polynomial shape multiplied by envelope function (in GaAs) $e^{-\sum_{i=1}^{N}{\left|z_{i}\right|}^{2}/4l_{B}^{2}}$. The generators of particular unitary cyclotron subgroups define the elementary exchanges of electron positions for various types of homotopy phases defined by invariants (Equation 4)—cf. Figure 6—and the multiparticle wave function of the corresponding FQHE ground state (at filling rate ν given by (Equation 5)) $\Psi_{\nu }\left(z_{1},\ldots ,z_{N}\right)$ must transform along the unitary representation of the braid describing the exchange of arguments of this function prescribed in accordance with the form of corresponding cyclotron group generators [36,37,38]. This defines the form of $\Psi_{\nu }\left(z_{1},\ldots ,z_{N}\right)$ in GaAs in an unambiguous manner [15]. Some examples of cyclotron braid subgroup generators are identified in Ref. [32] and will be discussed in the next paragraph in detail. The general form of the generators of cyclotron braid subgroups and the corresponding wave functions are given by Equations (Equation 8) and (Equation 11) (cf. next paragraph).

The construction of the trial wave function in the CF model is thus inaccurate because the projection onto LLL violates in an uncontrolled manner the symmetry of the multiparticle wave function, which must comply with the structure of the cyclotron subgroup generators corresponding to the particular homotopy invariant (Equation 4) and to scalar unitary representation of these generators (the violation of the symmetry in CF wave functions has been demonstrated in [39]). The unitary representation of the cyclotron braid subgroup must be a projective representation of the full covering braid group adjusted to original electrons, $\sigma_{i}\to e^{i\pi }$.

A simple example is the case of (Equation 4) with all $x_{i}=1$ for which (Equation 5) gives $\nu =\frac{1}{q}$ and the corresponding generators for the cyclotron subgroup [15] have the form $\sigma_{i}^{q}$, i=1,…,N−1 and $\sigma_{i}$ (i=1,…,N−1) and are generators of the full covering braid group describing exchanges of particles *i*-th and (i+1)-th ones. The scalar unitary representation of full braid group is of the form, $\sigma_{i}\to e^{i\alpha }$, α∈[0,2π), and for electrons must be taken as $\sigma_{i}\to e^{i\pi }$, i.e., α=π. In accordance with the form of the generators of cyclotron braid subgroup, $\sigma_{i}^{q}$ for this particular homotopy pattern (Equation 4), the projective scalar unitary representation must be, $\sigma_{i}^{q}\to e^{iq\pi }$. As proved in papers [36,37,38] the multiparticle wave function $\Psi (r_{1},\ldots ,r_{N})$ must transform along a scalar unitary representation of the braid if arguments of this function exchange their positions in the way prescribed by this braid. The representation $\sigma_{i}^{q}\to e^{iq\pi }$ together with the requirement that the multiparticle wave function in the LLL of interacting electrons on the plane must be holomorphic function, i.e., of polynomial form multiplied by the envelope $e^{-\sum_{i=1}^{N}{\left|z_{i}\right|}^{2}/4l_{B}^{2}}$ (the latter invariant to any action of braids), then one can rederive the Laughlin function (Equation 2) in an unequivocal manner.

The same holds for the simplest homotopy pattern (Equation 4) with q=1 and $x_{1}=1$, for which the scalar representation of related generators, $\sigma_{i}\to e^{i\pi }$, also unequivocally define the Laughlin function for q=1, which coincides in this case with the function (Equation 1). The Laughlin function for q=1 is the multiparticle function for *interacting* electrons on the plane and exposed to the magnetic field and in this exceptional case this function coincides with the Slater function of noninteracting 2D particles (Equation 1). The difference is, however, significant—the Laughlin function with q=1 describes the strongly correlated state of IQHE, whereas the Slater function (Equation 1) defines only the state of the gaseous system of *N* electrons at complete filling of the LLL, which is, however, not the same as IQHE. The homotopy phase corresponding to IQHE is the multiparticle quantum state of strongly correlated 2D electrons due to interaction, such a correlation is impossible in the gas without interaction. The Pauli correlations in the gas (which induce the form of Slater determinant (Equation 1)) are not induced by any interaction but only by the indistinguishability of identical particles.

More complicated invariants (Equation 4) and corresponding to them cyclotron braid group generators and multiparticle wave functions will be discussed in the following paragraph.

## 4. Cyclotron Braid Group Generators and Wave-Functions for Homotopy Phases

One can notice that frequently various commensurability patterns (Equation 4) can result in the same filling ratio (Equation 5). For instance $\nu =\frac{1}{3}={\left(1+1+1\right)}^{-1}={\left(\frac{1}{3}+\frac{1}{3}+\frac{1}{3}+1+1\right)}^{-1}$ and so on. Such redundant homotopy patterns at the same filling rate compete in energy and the ground state correspond to the pattern with minimal energy. This energy competition depends, however, on microscopic structure of material, though the homotopy patterns (Equation 4) are materially independent. The homotopy invariants unequivocally define the polynomial part of the multiparticle wave function in the LLL, the same in all Hall systems (and even in fractional Chern insulators when the magnetic field is substituted by the Berry field). The envelope functions (resistant to action of braid group elements) are in general different in various materials as defined by the microscopic structure of a particular material. This envelope function is interaction independent but its shape depends on the single-electron Hamiltonian including a crystal field, i.e., on all factors defining single-particle wave functions in a particular material. In GaAs it is conventionally assumed that single particle LLs are close to free electron states in magnetic field. Thus in GaAs the envelope of multiparticle wave function in the interacting system of $N\leq N_{0}$ electrons in the LLL is of the invariant form $e^{-\sum_{i=1}^{N}{\left|z_{i}\right|}^{2}/4l_{B}^{2}}$ (where $l_{B}=\sqrt{\frac{\hbar }{eB}}$ is the magnetic length and $N_{0}=\frac{BSe}{h}$ is the degeneracy of LLs, both quantities are interaction independent). This envelope function is the same in gaseous system, without any crystal field. Therefore this envelope does not favor any rank of neighbors in the Wigner crystal of electrons (the nearest neighbors or next-nearest ones).

Graphene is different; the single-particle LL states are distinct in comparison to free electrons, both in monolayer graphene [40] and in bilayer graphene [41]. Hence, in the graphene monolayer with identical homotopy invariants in the LLL (Equation 4), as in GaAs, the experimentally observed FQHE hierarchy is different. The difference is caused by distinct LLL single-particle wave functions in graphene, or more precisely, distinct envelope function depending in a different manner than in GaAs on magnetic field and on subband in the LLL in graphene. Four-fold quasi-degeneracy of LLs in the graphene monolayer [40] is caused by spin-valley structure and corresponds to ordinary spin-splitting (as in GaAs) and pseudo-spin splitting corresponding to two C atoms in elementary Bravais cells mixed with two nonequivalent Dirac points in corners of the hexagonal first Brillouin zone in graphene [40]. As was demonstrated experimentally [20] in graphene monolayer within a window of magnetic field in Landau fan diagram it has been noticed FQHE at $\nu =\pm \frac{1}{2},\;\pm \frac{1}{4},\;\pm \frac{3}{4}$ instead of the Hall metal state at these filling rates in GaAs and in the monolayer graphene outside this window (the magnetic field window depends on a sample and occurs at ca. 20 T and is of several T wide). Analogical FQHE states do not occur, however, in other subbands of the LLL in the graphene monolayer (i.e., at $\nu =\pm \frac{3}{2},\;\pm \frac{5}{2},\;\pm \frac{7}{4}$). This behavior has been explained [42] in the topological model. Note, that Hall metal can be accounted for by the limit y→∞ in (Equation 3), but the corresponding fractions with even denominators may also correspond to elements of the hierarchy (Equation 5) for the commensurability with next-nearest neighbors, e.g., $\nu =\frac{1}{2}={\left(1+\frac{1}{2}+\frac{1}{2}\right)}^{-1}$.

In GaAs no neighbors in the Wigner lattice are favored by the envelope function, and thus the correlations of closer electrons reduce the Coulomb repulsion energy. This corresponds to lower values of $x_{i}$. As loops are not distinguished, thus they can be ordered in the increasing way and for GaAs one can expect that more stable are phases with $x_{1}=x_{2}=\ldots =x_{q-1}=x$ and $x_{q}=y$, for which (Equation 5) attains the shape,
(6)$$\nu =\frac{yx}{(q-1)y\pm x},$$
which still is the generalization of (Equation 3) (CFs are reduced to the case x=1 only). One can verify that the formula (Equation 6) reproduces all observable in experiments fractions for FQHE in the LLL of GaAs [15,31,32].

### 4.1. Multiparticle Wave Functions for FQHE States in the LLL of GaAs

For identical indistinguishable particles their numeration is arbitrary in principle. Without any loss of generality one can consider that (i+1)-th particle is a nearest neighbor (in the sense of the Wigner crystal) of *i*-th one. It is sufficient to note that it would hold for any selected *i* and thus for all *N* particles as for each particle in the Wigner lattice exists its nearest neighbor. Due to the indistinguishability the problem of conventional numeration of electrons on the plane loses significance. The enumeration of indistinguishable electrons is not intuitive. The problems with enumeration of conventional distinguishable particles disappear here. Similar to next-nearest neighbors, for indistinguishable electrons only the integer rate $x_{\alpha }=\frac{N_{\alpha }}{N}$ of next-nearest neighbors is important, where $N_{\alpha }$ is the fraction of next-nearest neighbors, which create the Wigner sublattice of type α. In 2D there exist only two types of Wigner lattice—the hexagonal (of regular triangles) and regular (of squares), the latter energetically unstable. Thus, sublattices must also belong to these classes. For hexagonal structure $x_{\alpha }=3,4,7,9,\ldots$. Though the regular lattice is of higher energy than hexagonal one, its sublattice of next-nearest neighbor of first rank offers $x_{\alpha }=2$, not attainable in the hexagonal lattice case. This requires some rearrangement of hexagonal lattice but it is convenient energetically as the first rank next-nearest neighbors in hexagonal stronger reduces the electron subset to $x_{\alpha }=3$. Finally, $x_{\alpha }=2,3,4,7,9,\ldots$, cf. Appendix A.

The general homotopy invariant for cyclotron electron correlations of 2D electrons has the form given by Equation (Equation 4), where *q* is the number of loop of cyclotron orbit and $x_{i}$ indicates the fraction of next-nearest neighbors in Wigner lattice commensurate with *i*-th loop. The form of the invariant (Equation 4) results from the commensurability condition of singleloop cyclotron orbit with next-nearest neighbors of rank α, $\frac{BS}{N/x_{\alpha }}=\frac{h}{e}$, and from the form of the commensurability condition for *q*-loop cyclotron orbit with nearest neighbors only, which can be written as,
(7)$$\frac{BS}{N}=\frac{qh}{e}=\frac{h}{e}+\ldots +\frac{h}{e},$$
where the latter sum has *q* components. The signs ± in (Equation 4) indicate a possible inverted (−) or congruent (+) circulation of a loop with respect to the preceding one.

To the invariant (Equation 4) it corresponds the filling rate given by Equation (Equation 5). These filling rates are defined only by vectors $x_{1},\ldots ,x_{q}$, i.e., by homotopy patterns (Equation 4). The filling rate hierarchy of CFs (Equation 3) is the specific case of (Equation 4) and is related to (Equation 5) for $x_{1}=\ldots =x_{q-1}=1$, $x_{q}=y$ and ± before only the last term.

To the invariant (Equation 4) there correspond generators of a particular cyclotron subgroup in the following form,
(8)$$\begin{array}{c} b_{j}=\left(\sigma_{j}\sigma_{j+1}\ldots \sigma_{j+x_{1}-2}\sigma_{j+x_{1}-1}\sigma_{j+x_{1}-2}^{-1}\ldots \sigma_{j}^{-1}\right)\\ {\left(\sigma_{j}\sigma_{j+1}\ldots \sigma_{j+x_{2}-2}\sigma_{j+x_{2}-1}\sigma_{j+x_{2}-2}^{-1}\ldots \sigma_{j}^{-1}\right)}^{\pm 1}\\ \ldots \\ {\left(\sigma_{j}\sigma_{j+1}\ldots \sigma_{j+x_{q}-2}\sigma_{j+x_{q}-1}\sigma_{j+x_{q}-2}^{-1}\ldots \sigma_{j}^{-1}\right)}^{\pm 1},\\ j=1,\ldots ,N^{\prime }\;\;N^{\prime }=N-max\left(x_{i}\right), \end{array}$$
where the segment,
(9)$$\left(\sigma_{j}\sigma_{j+1}\ldots \sigma_{j+x_{i}-2}\sigma_{j+x_{i}-1}\sigma_{j+x_{i}-2}^{-1}\ldots \sigma_{j}^{-1}\right)$$
corresponds to exchange of the electron *j*-th with $j+x_{i}$-th one along the *i*-th loop of *q*-loop cyclotron orbit and $x_{i}\geq 1$ denotes here the fraction of nearest or next-nearest neighbors nested with this loop. For $x_{i}=1$ (the nearest neighbors) this whole segment (Equation 9) is $\sigma_{j}$.

The generators (Equation 8) define elementary exchanges of particles. Not all transpositions are possible—only those defined by the generators. Scalar unitary representations of generators (Equation 8) are $e^{i(1\pm 1\pm \ldots \pm 1)\pi }$ as for original electrons $\sigma_{j}\to e^{i\pi }$ and $\sigma_{j}^{-1}\to e^{-i\pi }$. Therefore, the segment (Equation 9) must induce the factor to the multiparticle wave function,
(10)$$\prod_{j=1,k=1;j<mod\left(j,x_{i},1\right)+\left(k-1\right)x_{i}}^{N^{\prime },N/x_{i}}\left(z_{j}-z_{mod\left(j,x_{i},1\right)+\left(k-1\right)x_{i}}\right),$$($N^{\prime }$ is the collection of admissible values of *j* at which the generator (Equation 8) can be defined, it is equal to $N-max(x_{i})$ for $x_{i}$ entering (Equation 8)) as the projective scalar unitary representation of this segment is $e^{i\pi }$ (or $e^{-i\pi }$ if it enters as inverted operator). In the above formula $mod(j,x_{i},1)$ is the rest of the division of *j* by $x_{i}$ with offset 1. Thus, the total multiparticle wave function corresponding to generators (Equation 8) acquires the form,
(11)$$\begin{array}{c} \Psi \left(z_{1},\ldots ,z_{N}\right)=A\prod_{j=1,k=1;j<mod\left(j,x_{1},1\right)+\left(k-1\right)x_{1}}^{N^{\prime },N/x_{1}}\left(z_{j}-z_{mod\left(j,x_{1},1\right)+\left(k-1\right)x_{1}}\right)\\ \times \prod_{j=1,k=1;j<mod\left(j,x_{2},1\right)+\left(k-1\right)x_{2}}^{N^{\prime },N/x_{2}}\left(z_{j}-z_{mod\left(j,x_{2},1\right)+\left(k-1\right)x_{2}}\right)\\ \times \ldots \\ \times \prod_{j=1,k=1;j<mod\left(j,x_{q},1\right)+\left(k-1\right)x_{q}}^{N^{\prime },N/x_{q}}\left(z_{j}-z_{mod\left(j,x_{q},1\right)+\left(k-1\right)x_{q}}\right)\\ \times e^{-i\sum_{i=1}^{N}{\left|z_{i}\right|}^{2}/4l_{B}^{2}}, \end{array}$$
for both two possibilities of scalar unitary representations related to ± in (Equation 4), causing only unimportant changes of sign.

One can notice that in the case of $x_{1}=x_{2}=\ldots =x_{q}=1$ and only + instead of ±, the Laughlin function (Equation 2) is reproduced. The envelope part of function (Equation 11), $e^{-i\sum_{i=1}^{N}{\left|z_{i}\right|}^{2}/4l_{B}^{2}}$, is correct only in GaAs (where the gaseous envelope is assumed) and this envelope changes in graphene according to the explicit form of single electron LL functions in graphene.

### 4.2. Simple Examples

It is instructive to write out explicitly some elementary examples of the function (Equation 11). For q=3, $x_{1}=x_{2}=1$ and $x_{3}=2$ the filling rate (Equation 5) is, $\nu ={\left(1+1+1/2\right)}^{-1}=2/5$, the generators are given by Equation (Equation 8) and the wave function by Equation (Equation 11). For N=4 this wave function has the explicit form,
(12)$$\begin{array}{c} \Psi \left(z_{1},z_{2},z_{3},z_{4}\right)=A{\left(z_{1}-z_{2}\right)}^{2}{\left(z_{1}-z_{3}\right)}^{3}{\left(z_{1}-z_{4}\right)}^{2}{\left(z_{2}-z_{3}\right)}^{2}{\left(z_{2}-z_{4}\right)}^{3}{\left(z_{3}-z_{4}\right)}^{2}\\ \times e^{-\sum_{i=1}^{4}{\left|z_{i}\right|}^{2}/4l_{B}^{2}}, \end{array}$$
which is apparently antisymmetric for admissible particle exchanges according to the generators (Equation 8), which in this case have the form,
(13)$$b_{j}=\sigma_{j}^{2}\sigma_{j}\sigma_{j+1}\sigma_{j}^{-1}=\sigma_{j}^{3}\sigma_{j+1}\sigma_{j}^{-1},$$
for j=1,2. These generators are illustrated in Figure 7. No other electron transpositions are admitted in this case, i.e., for homotopy invariant with q=3, $x_{1}=x_{2}=1$ and $x_{3}=2$ (at N=4).

For another example let us take $\nu =\frac{6}{11}={\left(1+1/2+1/3\right)}^{-1}$, i.e., q=3, $x_{1}=1$, $x_{2}=2$, $x_{3}=3$. From Equations (Equation 11) and (Equation 8) we get for N=6,
(14)$$\begin{array}{c} \Psi (z_{1},z_{2},z_{3},z_{4},z_{5},z_{6})=A\\ \times \left(z_{1}-z_{2}\right){\left(z_{1}-z_{3}\right)}^{2}{\left(z_{1}-z_{4}\right)}^{2}{\left(z_{1}-z_{5}\right)}^{2}\left(z_{1}-z_{6}\right)\\ \times \left(z_{2}-z_{3}\right){\left(z_{2}-z_{4}\right)}^{2}{\left(z_{2}-z_{5}\right)}^{2}{\left(z_{2}-z_{6}\right)}^{2}\\ \times \left(z_{3}-z_{4}\right){\left(z_{3}-z_{5}\right)}^{2}{\left(z_{3}-z_{6}\right)}^{2}\left(z_{4}-z_{5}\right){\left(z_{4}-z_{6}\right)}^{2}\left(z_{5}-z_{6}\right)\\ e^{-\sum_{i=1}^{6}{\left|z_{i}\right|}^{2}/4l_{B}^{2}} \end{array}$$
and related generators,
(15)$$b_{j}=\sigma_{j}\sigma_{j}\sigma_{j+1}\sigma_{j}^{-1}\sigma_{j}\sigma_{j+1}\sigma_{j+2}\sigma_{j+1}^{-1}\sigma_{j}^{-1}=\sigma_{j}^{2}\sigma_{j+1}^{2}\sigma_{j+2}\sigma_{j+1}^{-1}\sigma_{j}^{-1},$$
for j=1,2,3. Function (Equation 14) is apparently antisymmetric for admissible transpositions of electrons defined by generators (Equation 15). No other exchanges are possible at the filling rate $\nu =\frac{6}{11}$ for the homotopy invariant (Equation 4) with q=3, $x_{1}=1$, $x_{2}=2$ and $x_{3}=3$ at N=6 (cf. Figure 7). Note that polynomials in (Equation 12) and (Equation 14) are homogeneous as required.

For multiparticle wave functions (Equation 11) one can assess the energy. In order to assess the energy corresponding to multiparticle trial wave function the contribution to energy of mutual electron interaction as well as of interaction with positive jellium must be accounted for. For the disc geometry with the radius $r=\frac{R}{l_{B}}=\sqrt{\frac{2N}{\nu }}$ in units of $l_{B}=\sqrt{\frac{\hbar }{eB}}$ (where $\nu =\frac{N}{N_{0}}$, $N_{0}=\frac{BSe}{h}$ and $\rho =\frac{\nu }{2\pi }$—the density of electrons $\frac{N}{S}$ when *S* is expressed in $l_{B}^{2}$ units), the energy per single electron will contribute [43],
(16)$$\begin{array}{c} E_{jj}=\frac{\rho^{2}}{N}\int_{S}d^{2}r\int_{S}d^{2}r^{\prime }\frac{e^{2}}{4\pi \varepsilon_{0}\varepsilon \left|r-r^{\prime }\right|}=\frac{8}{3\pi }\sqrt{\frac{\nu N}{2}}\frac{e^{2}}{4\pi \varepsilon_{0}\varepsilon l_{B}},\\ E_{je}=-\frac{1}{N}<\Psi \left(r_{1},\ldots ,r_{N}\right)\left|\rho \int_{S}d^{2}r\sum_{i=1}^{N}\frac{e^{2}}{4\pi \varepsilon_{0}\varepsilon \left|r-r_{i}\right|}\right|\Psi \left(r_{1},\ldots ,r_{N}\right)>\\ =-\sqrt{2\nu N}<\Psi \left(r_{1},\ldots ,r_{N}\right)\left|\frac{1}{N}\sum_{i=1}^{N}F\left(r_{i}/r\right)\right|\Psi \left(r_{1},\ldots ,r_{N}\right)>\frac{e^{2}}{4\pi \varepsilon_{0}\varepsilon l_{B}},\\ E_{ee}=\frac{1}{N}<\Psi \left(r_{1},\ldots ,r_{N}\right)\left|\sum_{i<j}^{N}\frac{e^{2}}{4\pi \varepsilon_{0}\varepsilon \left|r_{i}-r_{j}\right|}\right|\Psi \left(r_{1},\ldots ,r_{N}\right)>, \end{array}$$
where $F\left(u\right)=\left\{\begin{array}{c} \frac{2E(u^{2})}{\pi },\;\;for\;u<1,\\ {}_{2}F_{1}\left(\frac{1}{2},\frac{1}{2};2;\frac{1}{u^{2}}\right),\;\;for\;u\geq 1, \end{array}\right.$ here E(x) is the complete elliptic integral, and ${}_{2}F_{1}\left(a,b;c;x\right)$ is the hypergeometric function ($\varepsilon_{0}$ is the dielectric constant, ε is the dielectric permittivity). $E_{jj}$, $E_{je}$, $E_{ee}$ are the energies of jellium-jellium, jellium-electron and electron-electron interaction, respectively (all calculated per single electron in the correlated state $\Psi (r_{1},\ldots ,r_{N})$). $E_{jj}$ is taken analytically (is independent of electron distribution), whereas $E_{je}$ and $E_{ee}$ can be estimated by the Metropolis Monte Carlo method of calculation of integrals with multi-argument wave functions. The activation energy in the state $\Psi (r_{1},\ldots ,r_{N})$ (per single particle and in units $\frac{e^{2}}{4\pi \varepsilon_{0}\varepsilon l_{B}}$) equals $E=E_{jj}+E_{je}+E_{ee}$. For the exemplary homotopy phases these energies are listed in Table 1 and Table 2.

From Table 1 and Table 2 we can notice that the activation energy grows with the increase of *N* in a similar manner as has been demonstrated for Laughlin functions [43]. Some other examples of various homotopy phases, their generators, wave functions and activation energies are presented in Ref. [32] (cf. also Table 3).

Some phenomenological modifications of Laughlin functions were studied [18,44] in order to account for possible anisotropic quantum Hall states. When the anisotropy is caused by striping [18], i.e., is of geometrical microscopic origin, then it can be handled easy within homotopy approach to electron correlations, but the molecular induced anisotropy [44] will need a more thorough topological discussion, which is out of the scope of the present paper.

## 5. Homotopy Invariants in Higher LLs and in Bilayer Hall Systems

Much more severe limits for the usability of the CF model than in the LLL occur in higher LLs. In all materials (GaAs, graphene monolayer and bilayer) the hierarchy of FQHE experimentally observed in higher LLs does not replicate the hierarchy form the LLL. This fact is not explained in the CF model. The distinction in FQHE hierarchy in higher LLs is linked with higher energy of LLs, which grows with the Landau index *n* as $E_{n}=\hbar \omega_{B}\left(n+\frac{1}{2}\right)$ where $\omega_{B}=\frac{eB}{m}$. Thus, for n=1 this energy is three-times larger than for n=0, or in general (2n+1)-times greater for arbitrary *n* than for n=0. This means that the smallest possible surface of the cyclotron orbit of electrons in *n*-th LL is equal to $\left(2n+1\right)\frac{h}{e}$, as the surface of cyclotron orbit is proportional to the energy of an electron.

Including a possible multiloop shape of cyclotron orbits (cf. Appendix B) their size is equal to $\left(2k+1\right)\left(2n+1\right)\frac{h}{e}$ in *n*-th LL and *k* is number of loops in an elementary braid. It is thus evident that in higher LLs (with n≥1) the size of the ordinary singleloop (with k=0) cyclotron orbits are large enough to match the closest electrons in the Wigner crystal of electrons from these LLs, contrary to the case n=0 when singleloop orbits were always too short. Hence, in the higher LLs we deal with a quite different type of FQHE in comparison to the LLL. Most prominent FQHE fractions in the first LL in GaAs, $\nu =2\left(3\right)+\frac{1}{3}$ and $\nu =2\left(3\right)+\frac{2}{3}$; in its two spin subbands are singleloop FQHE states, i.e., they correspond to topological invariants with nesting of cyclotron singleloop orbit to nearest and next-nearest neighbors in the Wigner lattice of electrons from these subbands (multiloop orbits and related CFs are not needed here). In the next LL with n=2 there occur more singleloop FQHE states, in GaAs at $4\left(5\right)+\frac{i}{5}$, where i=1,2,3,4. The same structure of FQHE hierarchy is repeated in graphene monolayer in all four spin-valley subbands for each LL (instead of only two spin subbands in GaAs). The relevant analysis in terms of topological invariants is presented in more detail in [45] in consistence with experimental observations in GaAs up to second LL [17] and similarly in graphene monolayer [19]. These states cannot be modeled with the help of CFs, because CFs always need multiloop structure of the cyclotron orbit with 2k+1 loops and 2k loops are represented as 2k quanta of fictitious magnetic field flux pinned to electrons. In higher LLs such a construction is meaningless as orbits are singleloop, but still the hallmark features of FQHE are present, i.e., $R_{xy}$ has plateaus at fractional ν, $\frac{h}{e^{2}\nu }$, assisted with minima of $R_{xx}$ at these fractions.

In bilayer Hall systems, like in bilayer graphene or in double GaAs 2DES, the model of CFs is also useless both in the LLL and in higher LLs. The reason for such a situation consists in the non-exact 2D topology of bilayer systems. If tunneling of electrons is admitted between layers, as in bilayer graphene or in two GaAs planes sufficiently closely located, then the braids and cyclotron orbits can be shared between two layers, which strongly affects the topological homotopy invariants. Two layers have independent surfaces and each of them contributes to the total flux of external magnetic field separately. Therefore, the topological invariants (like (Equation 4)) must refer to this new homotopy situation. For example, the three-loop cyclotron orbit can be distributed 2−1 or 1−2 between two layers, and in the cyclotron commensurability condition will take a part two loops instead of three. In this case, instead of FQHE at the fractional filling $\nu =\frac{1}{3}$, it should occur thus in the bilayer system FQHE at $\nu =\frac{1}{2}$, and actually such an unusual state is visible in the experiment in graphene bilayer [21] and in double GaAs 2D systems [24,25]. It is impossible to explain such a phenomenon in terms of CFs, but the homotopy theory is consistent with the experimental observations in all details [46,47,48]. A leakage of the magnetic field flux from one layer to the opposite one also strongly changes the FQHE hierarchy in higher LLs in bilayer graphene, which have been measured with record precision in whole Hall physics, up to the eighth LL subband and referred to as unconventional FQHE [22]. All the observed new features of FQHE from this experiment (ca. 30 various fractions out of CF model) can be derived within the homotopy invariant approach [48].

If one applies an electric field perpendicular to the bilayer Hall system, then the tunneling of electrons can be blocked in one direction (for graphene bilayer the blocking of interlayer electron hopping occurs at vertical voltage of 1−2 V). However, cyclotron loops are closed and must come back if they hop between layers; thus, their hopping between layers is completely blocked by the vertical voltage. This changes on demand the bilayer homotopy of trajectories to a monolayer one. Such an experiment has been performed with bilayer graphene [23] and it has been demonstrated that the change of FQHE hierarchy goes from that characteristic of a bilayer Hall system (e.g., the state at $\frac{1}{2}$) to a monolayer one (again the state $\frac{1}{3}$), when the vertical voltage is switched on. Another effect completely not understandable in CF terms is the disappearance of the FQHE state at $\nu =\frac{1}{2}$ in bilayer graphene if one substitutes the substrate of the graphene sample and in this way changes the order of subbands in the LLL of bilayer graphene. In this material we deal with eight-fold degeneracy of the LLL [41,49], the spin-valley four-fold degeneracy and additionally the so-called accidental degeneracy of n=0 and n=1 oscillator states (*n* is the Landau index) in bilayer graphene. Both states with n=0 and n=1 belong to the LLL in bilayer graphene due to specific its microscopic structure [41,49], but these state differ in the size of cyclotron orbits—in the subband with n=1 the cyclotron orbit is three-times greater than in the subband with n=0. If the latter subband is filled with electrons first, as in suspended sample of bilayer graphene, then the state $\frac{1}{2}$ is visible in the measurement [21]. However, if the former state (with n=1) is filled first, as in the case of hBN (hexagonal boron nitride) substrate, the $\frac{1}{2}$ FQHE state disappears in favor to $\frac{1}{3}$. This is clear in terms of the homotopy approach, because at Landau index n=1 three-loop orbits are not required because the singleloop ones are sufficiently large to reach nearest and next-nearest neighbors in the Wigner lattice of electrons—thus at $\nu =\frac{1}{3}$ FQHE corresponds in this case to a single-loop one not affected by the bilayer topology. This has been evidenced experimentally [50] and explained within homotopy approach [46].

Finally, let us mention the experiment [51,52] when the suspended scraping of the monolayer graphene initially in the insulating state converts into FQHE state (at the constant filling rate $\nu =\frac{1}{3}$) after short-time annealing by electric current impulse passing through the sample. This triggering on demand FQHE organization is out of reach for the CF model, but can be rationalized in terms of the braid homotopy approach [53]. An annealing reduces impurities and defects and increases the mobility of electrons in the sample. The mobility is proportional to the mean free path of carriers. If this mean free path is too short, the FQHE correlated state cannot be organized (all demonstrations of FQHE need mobility of order of 100,000 cm^2^/Vs or higher at which the mean free path of electrons exceeds the size of the sample). Thus, the mobility of electrons plays a triggering role in FQHE organization. This cannot be understood within the CF model. The homotopy approach finds, however, an argument that in order to organize the correlations of all electrons in the system at FQHE, the possibility of implementation of long braids is also required (besides the shorter ones limited by the cyclotron effect and electron repulsion). The long braids realize exchanges of electrons distantly located in the Wigner lattice and to implement such long braids (consisting of algebraic-group-multiplication of large number of generators) the sufficiently large mean free path of electrons is required (larger than the sample size). This explains the observation of the triggering of FQHE state organization in the graphene sample by its annealing [51,52,53].

## 6. Conclusions

We have demonstrated how to derive the popular model of CFs from first rules in the framework of the topological approach to multielectron correlations of 2D interacting electrons exposed to quantizing the perpendicular magnetic field. CFs turn out to be related with topological invariants protecting correlated phases of electrons at FQHE. The flux quanta of some auxiliary fictitious magnetic fields in construction of composite particles occur to be phenomenological representation of additional cyclotron loops of multiloop cyclotron orbits described upon the braid group approach. However, the CF model is possible to be formulated only in the case of simplest topological invariants when the commensurability of cyclotron orbits concerns the nearest neighbors in the Wigner lattice of electrons. This strongly limits the usability of the CF model and elucidates its failure in higher LLs in GaAs, and in graphene—in monolayer graphene in the LLL and in bilayer graphene in all its LLs. The limits of CF model identified in the present paper do not allow to explain in this model the experiments with FQHE observation in double Hall systems of GaAs, of the triggering of FQHE by annealing of graphene sample and the observations of so-called enigmatic FQHE states in the LLL of GaAs. All these unconventional FQHE phenomena require participation of correlations of next-nearest neighboring electrons in the Wigner lattice, beyond the CF model, or manipulation with loops of multiloop cyclotron orbits between layers of double-Hall-system, also impossible in the CF model formulation but understandable in terms of the homotopy braid group approach. Multiparticle wave functions proposed in the framework of CF model via projection onto LLL from higher LLs of auxiliary spinless Hall system are approximate and do not satisfy the braid symmetry requirements. The wave functions with proper braid symmetry can be constructed upon the topological homotopy approach. The CF model turns out to be, however, usable in the LLL of GaAs for FQHE states belonging to so-called Jain’s hierarchy. Moreover, fluxes attached to CFs can be described in terms of Chern-Simons gauge field theory with convenient calculation apparatus in Hamiltonian representation. 

## Figures and Tables

**Figure 1 materials-14-04267-f001:**
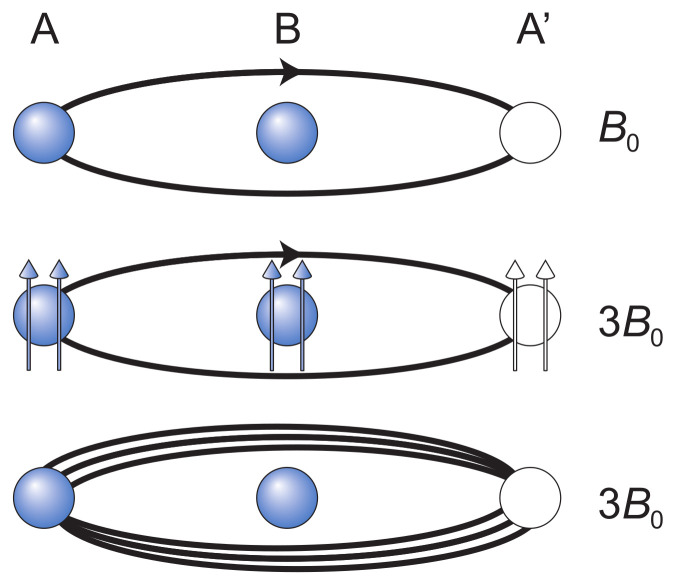
Two electrons A and B mutually interchange their positions on the plane at a magnetic field and a cyclotron orbit fits to the separation of nearest electrons in the Wigner lattice. The relative trajectory in magnetic field is shown—electron A traverses a relative cyclotron trajectory around electron B. Half of this trajectory, when the electron A takes the position A’, corresponds to a braid and the interchange of A and B electrons is completed—the phase shift of the wave function is π. In the upper panel the interchange of A and B electrons can be made along ordinary cyclotron orbit for field $B_{0}$, like in the case of IQHE. In the central panel, this interchange is proposed for CFs for FQHE at $\nu =\frac{1}{3}$ (for field $3B_{0}$), i.e., for electrons with two flux quanta of some fictitious field attached. The fictitious field reduces the external field $3B_{0}$ to $B_{0}$ and electrons A and B can interchange. The phase shift for the half of this trajectory is 3π, due to the Aharonov-Bohm-type effect. In the bottom panel the real structure of the cyclotron orbit and the braid (the half-piece of the cyclotron orbit) is shown for the external field $3B_{0}$—two additional loops are effectively modeled by two flux quanta added to CFs (as in the central panel). The phase shift for the transition described by the braid here is 3π according to the unitary representation of the braid. Three-loop cyclotron orbit is larger than singleloop which is too short at $3B_{0}$ and the three-loop orbit fits to the separation of nearest electrons in Wigner lattice (cf. Appendix B).

**Figure 2 materials-14-04267-f002:**
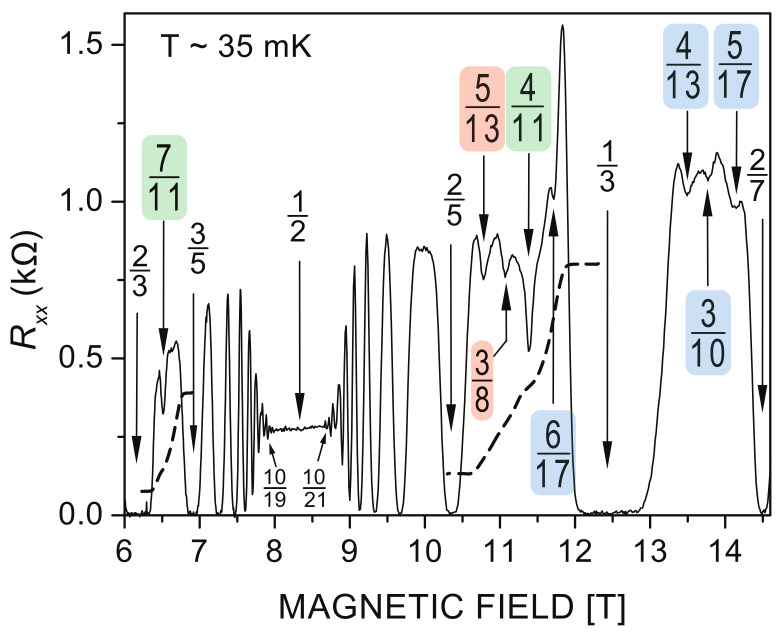
Longitudinal resistivity $R_{xx}$ at FQHE hierarchy in the fragment of the LLL of GaAs 2DES (after the experiment by Pan et al. [30]). Filling rates are indicated in colors for FQHE corresponding to so-called enigmatic states beyond the CF hierarchy (Equation 3). States marked with the same color are states at which the similar level of $R_{xx}$ local minima is achieved. The nonzero values of $R_{xx}$ evidence that not all electrons are involved in the correlated state, i.e., for filling rates given by (Equation 5) with $x_{i}>1$, which corresponds to correlations only of next-nearest neighbors in the Wigner lattice. Not correlated nearest neighbors contribute to the resistivity. Various portions of non-correlated electrons at different homotopy patterns (Equation 5) result in different residual $R_{xx}$.

**Figure 3 materials-14-04267-f003:**
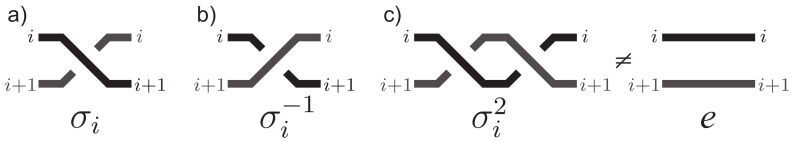
(**a**) Conventional geometric presentation for the generator $\sigma_{i}$ of the full braid group [34,35]—this generator describes the transposition of a particle *i*-th with (i+1)-th one on the plane $R^{2}$ when other particles remain on their positions. (**b**) Inverse braid $\sigma_{i}^{-1}$. (**c**) Square of generator $\sigma_{i}^{2}$, which for $M=R^{2}$ is not a neutral element of the group (though for $M=R^{3}$, $\sigma_{i}^{2}=e$ and this is a reason for the simplicity of the braid group in 3D in comparison to 2D case).

**Figure 4 materials-14-04267-f004:**
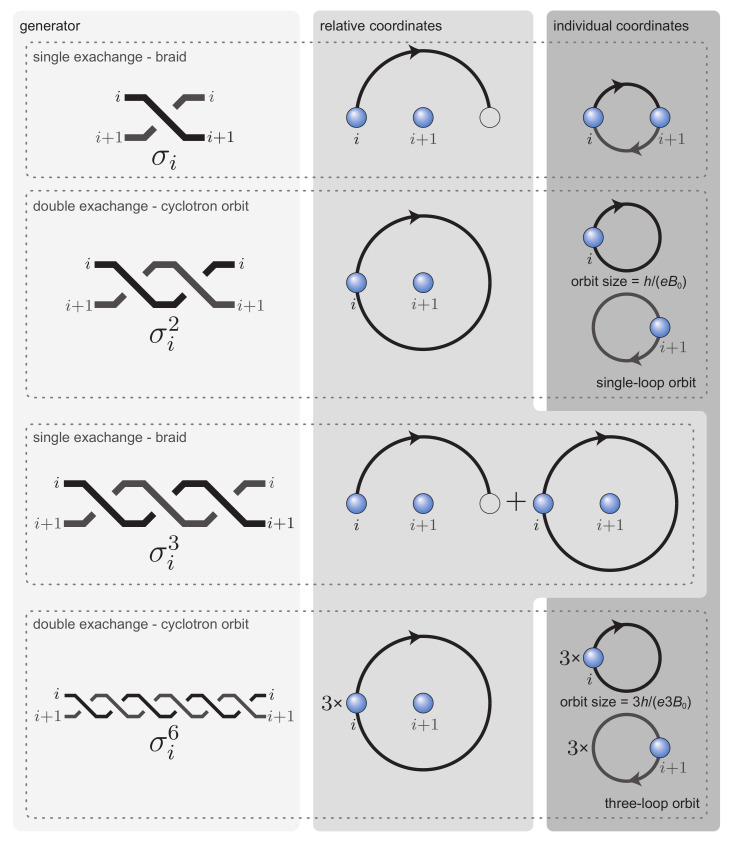
Schematic illustration that the elementary braids in the full group on the plane (the Artin group [34,35]), $\sigma_{i}$, and in its cyclotron subgroups, $\sigma_{i}^{q}$ (in the figure for q=3), must be half-pieces of cyclotron orbits. The illustration is shown in individual coordinates of particle pairs on the plane and in relative coordinates, both for the generators—braids and for the cyclotron orbits—square of braids (cyclotron braids are half-pieces of cyclotron orbits).

**Figure 5 materials-14-04267-f005:**
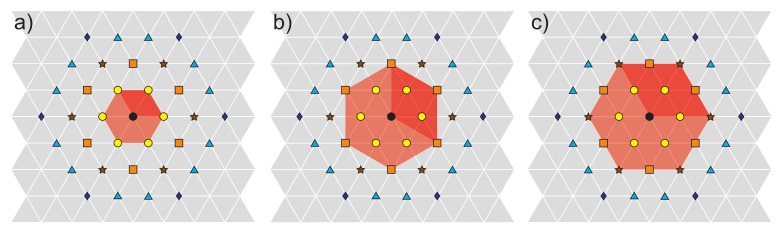
Triangle Wigner 2D lattice of repulsing electrons on a positive jellium at T=0 K with hexagonal Bravais elementary cell and indicated nearest neighbors (**a**) and two types of next-nearest ones (**b**,**c**). The hexagonal structure of sublattices of next-nearest neighbors is conserved.

**Figure 6 materials-14-04267-f006:**
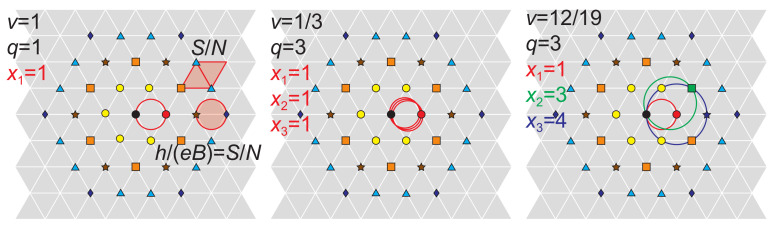
Illustration of some selected homotopy patterns protected by invariants (Equation 4).

**Figure 7 materials-14-04267-f007:**
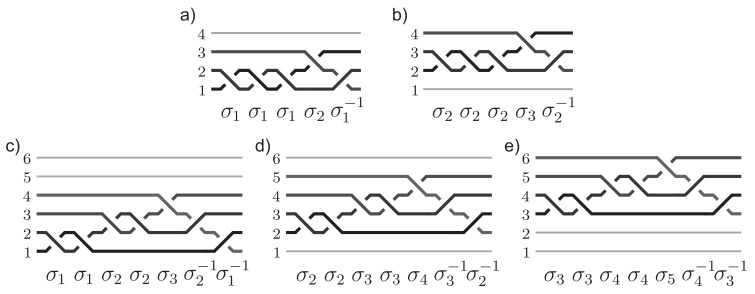
The generators for the homotopy patterns (1,1,2) at $\nu =\frac{2}{5}$ and N=4 (**a**,**b**) and (1,2,3) at $\nu =\frac{6}{11}$ and N=6 (**c**–**e**).

**Table 1 materials-14-04267-t001:** Activation energy for exemplary homotopy pattern $\left\{x_{i}\right\}=\left(1,1,2\right)$ for $\nu =\frac{2}{5}$ of FQHE correlations of CF type (with q=3 and $x_{1}=x_{2}=1$ and $x_{3}=2$).

Energy $\left[\frac{e^{2}}{4\pi \varepsilon_{0}\varepsilon l_{B}}\right]$	N=15	N=20	N=30	N=40	N=50
$E_{jj}$	−1.47021	−1.69765	−2.07919	−2.40084	−2.68422
$E_{je}$	−2.97161	−3.41054	−4.17117	−4.81280	−5.37656
$E_{ee}$	−1.07728	−1.28757	−1.66461	−1.98433	−2.26340
*E*	−0.424113	−0.425314	−0.427369	−0.427628	−0.428932

**Table 2 materials-14-04267-t002:** Activation energy for exemplary homotopy pattern $\left\{x_{i}\right\}=\left(1,2,3\right)$ for $\nu =\frac{6}{11}$ of FQHE correlations of not-CF type (with q=3 and $x_{1}$, $x_{2}=2$ and $x_{3}=3$).

Energy $\left[\frac{e^{2}}{4\pi \varepsilon_{0}\varepsilon l_{B}}\right]$	N=15	N=20	N=30	N=40	N=50
$E_{jj}$	−1.71684	−1.98243	−2.42797	−2.80358	−3.13450
$E_{je}$	−3.48480	−4.01557	−4.87400	−5.63216	−6.30237
$E_{ee}$	−1.30383	−1.56674	−1.97696	−2.35739	−2.69621
*E*	−0.464135	−0.466401	−0.469062	−0.471188	−0.471661

**Table 3 materials-14-04267-t003:** Comparison of energy values (per particle in units, $\frac{e^{2}}{4\pi \varepsilon_{0}\varepsilon l_{B}}$) obtained by exact diagonalization (Ex. diag.) and by quantum Monte Carlo simulation (MMC sim.) for few exemplary filling fractions in GaAs with FQHE corresponding to patterns q,x,y (cf. Equation (Equation 6)) (Metropolis Monte Carlo simulation for the proposed topology-based wave functions for 200 particles [32]).

$\nu =N/N_{0}$	3/7	4/9	5/11	2/9	3/13	4/17
q,x,y	3, 1, 3	3, 1, 4	3, 1, 5	5, 1, 2	5, 1, 3	5, 1, 4
**MMC sim.**	−0.441974	−0.446474	−0.451056	−0.342379	−0.348134	−0.351857
**Ex. diag.**	−0.442281	−0.447442	−0.450797	−0.342742	−0.348349	−0.351189

## Data Availability

The author confirms that the data supporting the findings of this study are available within the article.

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
