# Peer review of "Limits of Applicability of the Composite Fermion Model"

_materials, 2021, doi:10.3390/ma14154267_

Round 1

Reviewer 1 Report

The manuscript contains a critics of the composite fermion model (CFM) to explain the fractional quantum Hall effect (FQHE) (as it was published by Jain, PRL 63, 199 (1989)). Alternatively, the complete hierarchy of FQHE (including the so-called enigmatic FQHE states which were experimentally observed but cannot be explained by the CFM) is explained by a topological approach developed by the author and previously published in several publications.

The idea of the author is rather original and interesting. It explains the complete hierarchy of FQHE states in a quite natural way. The theory is explained in an understandable way. However, the present manuscript contains no new result in comparison to the publication of the same author which appeared in 2018 in Physical Review A 97, 012108 with the title: "Application of path-integral quantization to indistinguishable particle systems topologically confined by a magnetic field". For instance

Table I in PRA is identical to Table 1 here,
Fig. 3 in PRA combines Figs. 2 and 7 of the new manuscript,
Eq.(7) in PRA is identical to Eq. (7) here,
Eq. (6) in PRA is identical to Eq. (2) here, and so on.

I could not observe any new calculation, or any new idea in the present manuscript in comparison to previous publications. The text is different to previous publications, it is now much more based on a critics of the CFM than it was in the PRA publication. But already there, this critics is clearly expressed. So, after Eq.(4) of the PRA paper it is written that "for x>1 the hierarchy (4) is beyond the possibility of Jain's model of CF."

The chapter 4 of the present manuscript "Homotopy invariants in higher LLs and bilayer Hall systems" contains an application of the author's topological approach to the FQHE of higher Landau levels or to bilayer graphene. Also, these results were previously published by the same Autor in J Phys. Condensed Matter 31 475601 (2019). 

It should be remarked that, besides these two publications mentioned above, the same author published his topological approach in 5 other journals and in co-authorship with a second author in 4 further papers.

So, I am sorry, I cannot see any reason to publish the present manuscript in a scientific journal since it repeats already published results and a publication would contradict the basic rules of scientific publications.

If the author had the intention to write a review article, that should be clearly mentioned in the beginning. But, to my mind, a review which is just a summary of the works of one only scientist is not justified. It should contain all the relevant works of a certain scientific field which is not the case of the present manuscript.

Author Response

Dear Referee,

Thank you very much for a very thorough and very inquisitive review of our submission. You are suggesting that it cannot be published because some (small part actually) of the equations and figures are similar to those posted in the PRA (Phys. Rev. A 97, 012108 (2018) by J. Jacak, Application of path-integral quantization to indistinguishable particle systems topologically confined by a magnetic field – please note, however, that this paper was addressed to quite different issue, not to CFs). To avoid such an argumentation, we completely revised the relevant  part of the present manuscript (we have removed previous examples with equations and the figure and table) and now we have replaced them with new calculations and examples (in color in the resubmission for easy comparison). The new Tabs 1 and 2 with new data are included and  Fig. 7 is completely changed.  Thus, your most important objection has been removed and we agree with you that such a corrections have increased the clarity of the discussions and explanation, on the other hand.

Moreover, it should be emphasized that your other comments are not truly scientific, but rather  opposed to the new concept and any criticism directed at composite fermions. Actually, the irrelevance of the CF models was previously only mentioned in our previous publications (as you have cited, in only single sentence from the PRA). Nevertheless, the problem is of a great importance and it certainly deserves a broader and complete discussion as presented in the submission. This is related to the role of FQHE in condensed matter and, in particular, of the CF model. The CF model was honored with the Buckley Condensed Matter Prize for Jain in 2002  (the founder of CF model), and even (in the last 10 years) the model has been repeatedly submitted to NP committee  (by the lobby of das Sarma, Chakraborty  or other CF model supporters). Several books have been published about CFs (by Jain himself and by other authors), and hundreds or more of publications about CFs appeared in journals (still publishing despite the model basis occurred wrong and current experiments precluded the CF model in new materials). This is why a precise explanation of the applicability of the CF model is so important. We do not exclude the CF model, but rather explicitly derive it (in a specific and limited situation when it is correct) from more fundamental premises within the topological approach. You yourself agree that our results are sound, but you try to make it difficult to publish them. This is unacceptable in the scientific community. A high-level independent journal such a Materials MDPI is a very good  platform (additionally  Open Access) for presenting the relevant discussion, in particular in targeted Special Issue of Materials. Please accept that explaining the applicability of the popular CF model is really a very important issue.

So instead of anonymous degradation of our efforts, we (as one of the Guest Editors for Special Issue of Materials, “Topological Approaches to 2D Multielectron Correlated States”) kindly invite you to contribute with a polemic submission to this SI. From your report is visible that you are very well oriented in the CF model as well as  in our approach (in all details from our former publications). Your contribution and the public discussion about  controversies and your objections in the SI of Materials  would be thus of great importance and value and strongly desired.

On the other hand we hope that we have improved the presentation in the resubmission  according also to your constructive recommendations (we removed to practically zero the overlap with our former publications, and substitute formerly presented examples with new ones with more detailed calculations and explanations).

So please kindly accept our invitation to contribute to the SI with your comments and controversies specially addressed to CF model which is apparently close to your expertise.

Sincerely yours,
J. E. Jacak

Reviewer 2 Report

The author discusses the limits of the composite fermion theory when such a theory is applied, in particular, to certain “enigmatic” states in the lowest Landau Level (LLL) (the likes of filling factor 4/11, 5/13, …), fractional Hall states in higher LLs in GaAs and in graphene. The work is interesting and well written. I have the following comments:

A) Page 1, After they mention anyons and references [5,6] they should also cite a specific work that is relevant to anyons and deals with a general anyon wave functions for such systems:

  • Phys.: Condens. Matter 17, 2977 (2005).

B) Page 2, they mention “the fractional Hall hierarchy in higher LLs in GaAs [16]”. They must follow up the reference [16] with few other references as below:

  • Appl. Phys. 107, 09C504 (2010).

C) Page 12, Table 1, some comment is needed on what is the number of the particles used in the diagonalizations (I expect something of the order 10-20!). Comment is also needed on explaining the (expected) differences of comparing exact diagonalization energy for 10-20 particles to the Monte Carlo ones (where 200 particles are used).

D) The author argues that the composite fermion (CF) theory can be inadequate for several scenarios. I find these arguments articulated well. Therefore, I would tend to believe that the CF theory may also have some limits on explaining anisotropic quantum Hall states that break the rotational symmetry of the system like those in the references below:

  • AIP Adv. 7, 055804 (2017)

A very short discussion on this matter would be most welcomed.

I would recommend publication of the manuscript after the authors amend the manuscript to add the new references listed above and address the comments in the points listed above.

Author Response

Dear Referee,

Thank you very much for the assessment of our submission and  for the positive opinion. We agree with all your recommendations to improve the presentation. According to your suggestions we have added new references. We added also a small comment addressed to the issue of anisotropic case, as you suggested in the report. Moreover, we have made the change of examples and explicit calculations to avoid any overlap with our previous publications. The dependence of number of electrons in numerical simulations is explicitly shown in new Tabs 1 and 2. Fig. 7 is also changed.  All corrections are marked in red color in the resubmission PDF. We hope that we have improved the presentation in a satisfactory way.  A linguistic corrections and proofreading of the text were also made.

Once more we express our great gratitude for your very nice opinion. 

Sincerely yours,
J. E. Jacak

Round 2

Reviewer 1 Report

  Referee report of the revised version

Author:  Janusz E. Jacak 

Title: Limits for composite fermion model applicability

Journal: Materials

Publisher: MDPI

The author added new material in the revised version of the manuscript such that it contains now indeed scientific results which were not published before. But, even if the manuscript contains now some formerly unpublished results and fulfils in such a way the minimal requirement of a scientific publication, its quality remains poor. The main critics are the following:

(A) It is a good idea to include the convergence of the energy with particle number in the Metropolis Monte Carlo simulation (MMC sim.). But the author now deleted completely the results with 200 electrons and the comparison with exact diagonalisation which was present in the first version of the manuscript. This comparison is interesting and should be re-inserted.

(B) Apparently, the author was in a hurry to improve the manuscript since by including new material he did several mistakes. For instance, there is a mistake in line 2 of Eq(10) : x1 has to be replaced by x2. Then, on line 275 "neighbor ... of t-th one" should probably read "neighbor .... of i-th one".

(C) What are the conclusion out of the numerical results? Can these numerical results be compared with other proposed wave functions?

(D) The linguistic quality of the manuscript remains poor. It starts already with the title which should be better expressed as "Limits of applicability of the composite fermion model". Two further examples are (i) the first sentence of the Abstract which is too long and should be splitted, and (ii) line 103 where ",in the spin-down its subband" should be replaced by ",in its spin-down subband". Many other examples could be given.
(Line 280: "Similarly with next-nearest neighbours. " is not a sentence.)

(E) The representation is very redundant with many repetitions. For instance equations (4) and (5) are repeated in equations (7) and (9).

(F) The author critics very strongly the composite fermion model. Some of his critics is eventually justified, but the composite fermion has also advantages. Especially the connection with field theoretical models (Chern-Simons field theory) is possible for the CFM. So, the invent of the CFM opened many perspectives which go even beyond the FQHE. The author is not mentioning these positive arguments for the CFM at all. Therefore, I find the present manuscript not well balanced, arguing only against the CFM.

To my mind, the manuscript should be completely rewritten in avoiding the repetitions mentioned above (E), concentrating on the explicit construction of the wave functions and concentrating on the conclusions out of the numerical results. Care should be taken to compare with numerical results for other wave functions. Also the critics of the CFM should be rewritten in giving a more balanced presentation.

The positive ideas of the present manuscript are the explicit construction of a wave function to explain the so-called enigmatic states of the FQHE which go beyond the CFM hierarchy and the numerical results. In rewriting the manuscript, the author should concentrate on these points.  

Author Response

Dear Referee,

Thank you again for the inquisitive review and helpful recommendations. We have corrected the new resubmission along lines indicated in the report 2 of the  Referee #1. Some explanations are additionally listed below.

Ad (A)
We have placed in the paper again the results of Metropolis Monte Carlo simulations for 200 electrons.  The re-inserted table contains the comparison with exact diagonalization.

Ad (B)
Absolutely right.  The misprints made in hurry at first resubmission are now corrected. Thank you for identification of them.

Ad (C)
As written in text, the wave functions were previously not known for fractions out of Jain’s CF hierarchy, and, moreover, for fractions from this hierarchy were written with the help of projection onto the LLL from higher LLs and have a form not satisfying braid symmetry requirements. As also noted in the submission, the average energies over trial wave functions upon the CF model give relatively good activation energies for FQHE phases, as the projection onto the LLL plays the role of variational procedure. Moreover,  we have verified  by Metropolis Monte Carlo simulations that small changes in the form of trial multielectron wave functions (including those breaking the  braid symmetry, like in CF model) cause only minor shifts of mean energy.

Ad (D)
We agree with this critique. We corrected the indicated phrases including the suggested reformulation of the title.

Ad (E)
We have removed the redundancy in the presentation including repetitions of  formulae.

Ad (F)
The Referee is the advocate of the CF model. We repeat our kind invitation addressed to the Referee, to present his/her point of view as the contribution to the Special Issue of Materials “Topological Approaches to 2D Multielectron Correlated States”. It would be excellent to compare various approaches on this platform.
With regard to the linkage of CFs with Chern-Simons gauge field theory, the short comment is now added in the new resubmission (in blue color in the PDF file).

“CF model turns out, however, usable in the LLL of GaAs for FQHE states belonging to so-called Jain’s hierarchy. Moreover, fluxes attached to CFs can be described in terms of Chern-Simons gauge field theory with convenient calculation apparatus in Hamiltonian representation.”

To clarify this linkage it is crucial to note that Chern-Simons gauge field is not a canonical transformation – in fact it is not a true gauge (in contrast to e.g., Maxwell gauge transformations), because it does not conserve the quantum statistics, and changes if for demand (by fixing the parameter of this transformation). This choice of the parameter is arbitrary and not supported by any derivation, it is similarly artificial as the choice of number of flux quanta of the auxiliary fictitious magnetic field pinned to CFs.  Hence, Chern-Simons gauge field theory applied to FQHE is also only illustration (not derivation!) of CFs, which were  phenomenological illustration  of topological invariants. Because the Chern-Simons gauge field theory needs rather extensive presentation towards such a discussion, thus this issue is beyond the scope of the present paper and deserves  another separate publication. We plan to release such a paper.

In Conclusions we have now added more explicit comments in the manner  recommended by the Referee #1 (in blue color in PDF file).  Yes, we agree that the construction (by Eqs (8) and (11)) of wave functions in an analytic form is the progress in the FQHE theory. Such functions were previously given in an accurate manner by Laughlin only for 1/q hierarchy. These Laughlin functions we rederived in our approach  with a mathematical rigor. This  is a verification of correctness of the braid homotopy approach. No other theory of FQHE was able to derive the Laughlin functions. They were only postulated by Laughlin, not derived. For other filling rates (apart of  the fundamental hierarchy 1/q)  the trial wave functions were only proposed approximately, including those given by Jain in his theory of CFs (via projection from higher LLs [9]) or by Halperin in the form of the multicomponent generalizations of Laughlin functions [8] (references in submission). We propose the wave functions generated by the braid symmetry in the unambiguous way in the LLL. It is interesting to note once more that the variational procedures with CFs wave functions and multicomponent ones, on the other hand, result in the similar mean energy despite different analytical forms of trial wave functions (also not far from the mean energy assessed for braid symmetry induced functions). This is caused by the structure of the energy, which consists of the jellum-jellium, jellium-electron and electron-electron interaction energies (per one electron) in the multielectron system, which varies with number of electrons. We have illustrated this in Tables 1 and 2. For large number of electrons the  sum of these three energy components saturates (in the thermodynamic limit) revealing the predominance of jellium-electron (negative) over electron-electron and jelium-jelium energy components (both positive). As the jellium-jellium energy does not depend on electron distribution, the activation energy mostly depends on the electron-electron interaction. The minimization of this interaction is thus crucial and is achieved by multiple zeros in the wave function (stronger separation of electron densities), as in Laughlin functions or in each variant of multiparticle function of polynomial shape. For large number of variables, small changes in this polynomial do not cause a significant difference in integrals.

We hope that we corrected the manuscript in a satisfactory manner.

Sincerely yours,
Janusz E. Jacak

Reviewer 2 Report

The author has done a good job to amend the manuscript by considering the comments in the Reviewer’s report. I recommend publication.

Author Response

Dear Referee,

Thank you again for the very helpful comments and recommendations. We are greatly indebted for a positive opinion.

Sincerely yours
Janusz E. Jacak